# Intraoperative Management of Parathyroid Glands and Long-Term Outcome of Parathyroid Function Following Total Thyroidectomy

**DOI:** 10.3390/diagnostics15050593

**Published:** 2025-02-28

**Authors:** Feng-Yu Chiang, Kang Dae Lee, Kyung Tae, Kwang Yoon Jung, Chih-Chun Wang, Tzer-Zen Hwang, Che-Wei Wu, Shih-Wei Wang, Yu-Chen Shih, Tzu-Yen Huang

**Affiliations:** 1Department of Otolaryngology-Head and Neck Surgery, E-Da Hospital, Kaohsiung 824, Taiwan; fychiang@kmu.edu.tw (F.-Y.C.); ccw5969@yahoo.com.tw (C.-C.W.); tzhwang@hotmail.com.tw (T.-Z.H.); 2School of Medicine, College of Medicine, I-Shou University, Kaohsiung 824, Taiwan; 3Department of Otolaryngology-Head and Neck Surgery, College of Medicine, Kosin University, Busan 49267, Republic of Korea; kdlee59@gmail.com; 4Department of Otolaryngology-Head and Neck Surgery, College of Medicine, Hanyang University, Seoul 04763, Republic of Korea; kytae@hanyang.ac.kr; 5Department of Otolaryngology-Head and Neck Surgery, Korea University College of Medicine, Seoul 02841, Republic of Korea; kyjungmd@gmail.com; 6Department of Otolaryngology-Head and Neck Surgery, Kaohsiung Medical University Hospital, Faculty of Medicine, College of Medicine, Kaohsiung Medical University, Kaohsiung 807, Taiwan; cwwu@kmu.edu.tw (C.-W.W.); wanglipid906@gmail.com (S.-W.W.); 7Department of Otolaryngology-Head and Neck Surgery, Kaohsiung Municipal Siaogang Hospital, Kaohsiung Medical University Hospital, Kaohsiung Medical University, Kaohsiung 812, Taiwan; 8Department of Otolaryngology, E-Da Cancer Hospital, I-Shou University, Kaohsiung 824, Taiwan; 9Department of Otorhinolaryngology, School of Post-Baccalaureate Medicine and School of Medicine, College of Medicine, Kaohsiung Medical University, Kaohsiung 807, Taiwan; 10Department of Otolaryngology-Head and Neck Surgery, Kaohsiung Medical University Gangshan Hospital, Kaohsiung Medical University Hospital, Faculty of Medicine, College of Medicine, Kaohsiung Medical University, Kaohsiung 820, Taiwan

**Keywords:** parathyroid glands, total thyroidectomy, stabbing test, hypoparathyroidism, parathyroid in situ preservation, parathyroid autotransplantation

## Abstract

**Background/Objectives:** In situ preservation is the primary strategy to preserve parathyroid gland (PG) function during thyroid surgery, while autotransplantation is used when inadvertent removal or devascularization occurs. Deciding on the optimal approach intraoperatively for exposed PGs remains challenging. This study evaluates intraoperative PG management strategies and long-term outcomes of PG function following total thyroidectomy. **Methods:** This retrospective study included 543 patients undergoing primary total thyroidectomy, excluding those with comorbid parathyroid disease. A stabbing test assessed the vascular supply of exposed PGs. PGs with fresh blood oozing after the test were preserved in situ; otherwise, they were autotransplanted. Intact parathyroid hormone (iPTH) and ionized calcium (iCa) were measured preoperatively and on postoperative day 1 (PO-1D), and during follow-up. Permanent hypoparathyroidism (PHPS) was defined as iPTH < 15 pg/mL, iCa < 4.2 mg/dL, or continued need for calcitriol or calcium supplementation after a postoperative period of 12 months (PO-12M). The PHPS rate was compared with the corresponding intraoperative PG status. **Results:** A total of 528 patients were enrolled in this study. At PO-1D, 434 patients (82.2%) had iPTH ≥ 15 pg/mL, 65 (12.3%) had iPTH between 4 and 15 pg/mL, and 29 (5.5%) had iPTH < 4 pg/mL. At PO-12M, 527 patients (99.81%) had iPTH ≥ 15 pg/mL, 1 (0.19%) had iPTH between 4 and 15 pg/mL, and none had iPTH < 4 pg/mL. Five patients (0.95%) were in PHPS after PO-12M. Among the 462 patients with at least one viable PG preserved in situ, the PHPS rate was 0.2%, compared to 6.1% (66 patients) for those without a viable PG preserved in situ (*p* < 0.001). **Conclusions:** Permanent hypoparathyroidism is rare when at least one viable PG is preserved in situ during total thyroidectomy. The stabbing test is a simple, useful, and cost-effective method to assess the vascular supply of exposed PGs, providing surgeons with essential information for intraoperative PG management.

## 1. Introduction

Total thyroidectomy (TT) is a common surgical procedure used to treat patients with thyroid cancer, large symptomatic multiple goiter, and poorly controlled Graves’ disease. However, the occurrence of permanent hypoparathyroidism after TT is not uncommon. In some reports with higher prevalence rates, Juan J Díezet et al. [1] reported a 14.5% rate of permanent hypoparathyroidism in a multicenter and nationwide retrospective analysis. When TT was performed in a low-volume institution, the incidence of permanent hypoparathyroidism was up to 31.5% [2]. In a meta-analysis published in 2024 [3], a selection of studies from 2021 to 2024 showed that the incidence of transient hypoparathyroidism after TT ranged from 2.3% to 54.5%, while the incidence of permanent hypoparathyroidism ranged from 1.7% to 21.9%. Although some meta-analyses report a lower incidence of permanent hypoparathyroidism, the definition of hypoparathyroidism varies from study to study. Nevertheless, this complication results in the need for long-term use of calcitriol or calcium supplements, which not only increases healthcare costs, but also adversely affects quality of life.

In situ preservation and autotransplantation are common methods used to preserve parathyroid gland (PG) function in TT. Over the past few decades, PG autotransplantation gained popularity and was even routinely used because many studies [4,5,6,7,8,9,10,11,12,13] had reported that it was an effective procedure to reduce the incidence of permanent hypoparathyroidism after TT. However, the benefit of routine PG autotransplantation has been questioned. Numerous studies [1,14,15,16,17,18,19] have shown that a higher number of PG autotransplantation not only increases the rate of temporary postoperative hypocalcemia, but also increases the rate of permanent hypoparathyroidism due to inadequate functional recovery of autotransplanted PGs. Therefore, most thyroid surgeons now accept the strategy that in situ preservation of well-vascularized PGs and autotransplantation of devascularized PGs is the best strategy to achieve a good outcome of PG function preservation.

However, during surgery, it can be difficult to assess vascular supply of exposed PGs by visualization. A well-colored PG does not necessarily have vascular supply and therefore the in situ preserved PG may not survive [20,21,22]. Conversely, a discolored PG does not imply a complete loss of vascular supply and may be mistakenly autotransplanted [23]. In recent years, indocyanine green (ICG) angiography has been reported to be useful in the assessment of PG vascular supply by measuring fluorescence intensity [24,25,26,27]. However, equipment cost, intraoperative time consumption, potential side effects of ICG, lack of standards for the interpretation of fluorescence results, and unclear benefits for preventing permanent hypoparathyroidism limit its use for most thyroid surgeons [28,29,30].

To improve intraoperative decision-making, various techniques have been proposed to assess PG viability. In this study, we investigated intraoperative PG management strategies and the long-term outcomes of PG function in TT patients.

## 2. Materials and Methods

### 2.1. Patients

From March 2020 to October 2023, 543 patients underwent primary TT by a single surgeon (F-Y.C.). Fifteen patients were excluded due to comorbid parathyroid disease (ten patients with primary parathyroid adenoma and five patients with renal hyperparathyroidism). No patient in this study showed preoperative hypercalcemia, bone disease, or nutritional or other rickets. Indications for TT included thyroid cancer, high suspicion of malignant thyroid nodules, large symptomatic multiple nodules, and poorly controlled Graves’ disease. Computed tomography (CT) scans were routinely performed in all patients to evaluate paratracheal and lateral cervical lymph nodes. All surgeries utilized intraoperative nerve monitoring (Medtronic NIM System, Jacksonville, FL, USA) and an energy-based device (LigaSure Exact Dissector, Medtronic, Mineapolis, MN, USA).

### 2.2. Surgical Procedures and Intraoperative PG Management

All patients underwent open thyroidectomy with the same standard surgical procedure: (1) first, excision of the pyramidal lobe and anterior laryngeal lymph nodes; (2) then, dissection of the upper pole of the thyroid gland along the capsule; (3) subsequently, dissection of the lateral thyroid lobe and recurrent laryngeal nerve; and (4) finally, central neck dissection (CND) or lateral neck dissection (LND) when metastatic lymph nodes were suspected on ultrasonography or CT scan. The same strategies for PG preservation were used intraoperatively: (1) careful inspection of the thyroid capsule to identify PG and meticulous dissection of the thyroid capsule to preserve the PG and vascular supply; (2) routine assessment of the vascular supply for each exposed PG with a stabbing test (i.e., PG is stabbed with a 23G needle); (3) the PG was considered viable and preserved in situ if fresh blood oozing was found after fine needle stabbing (positive stabbing test, Figure 1A–C); and (4) the PG was considered non-viable and autotransplanted into the muscle pocket if no fresh blood oozing was found after repeated PG stabbing or no fresh blood oozing after bleeding from old congested blood (negative stabbing test, Figure 2A–C). Data were recorded and registered for each visible PG, including location, stabbing test results, and in situ preservation or autotransplantation. Patients were divided into four intraoperative PG management status groups: group 1, at least one viable PG preserved in situ and at least one PG autotransplanted; group 2, at least one viable PG preserved in situ, but no PG autotransplanted; group 3, no viable PG preserved in situ, but at least one PG autotransplanted; and group 4, no visible PG preserved in situ or autotransplanted.

### 2.3. Measurements of iPTH and iCa

None of the patients received preoperative calcitriol or calcium supplements. Intact parathyroid hormone (iPTH) and ionized calcium (iCa) levels were measured preoperatively and on postoperative day 1 (PO-1D, i.e., 12–24 h after surgery) in all patients. In this study, the normal range of iCa levels is defined as the preoperative iCa mean ± 2SD, thus hypocalcemia is defined as iCa < 4.2 mg/dL. Hypoparathyroidism is defined as iPTH < 15 pg/mL.

When iPTH was less than 15 pg/mL or iCa was less than 4.2 mg/dL at PO-1D, long-term monitoring of iPTH and iCa was performed postoperatively at approximately 2 weeks (PO-2W), 2 months (PO-2M), 6 months (PO-6M), and 12 months (PO-12M). The measurement was discontinued when iPTH levels ≥ 15 pg/mL and iCa ≥ 4.2 mg/dL after withdrawing calcitriol and calcium supplements. Patients with iPTH ≥ 15 pg/mL and iCa ≥ 4.2 mg/dL at PO-1D no longer required calcium measurement, calcitriol, or calcium supplementation. Patients with iPTH between 4 and <15 pg/mL or iCa between 4.0 and < 4.2 mg/dL were routinely given calcium carbonate (0.5 to 1 g, bid) and calcitriol (0.5 to 1.0 mg, bid). Additional intravenous calcium chloride was given to patients with iPTH < 4 pg/mL or iCa < 4.0 mg/dL or symptoms of hypocalcemia.

Permanent hypoparathyroidism status (PHPS) was defined as the presence of any of the following after PO- 12M: (1) iPTH < 15 pg/mL, (2) iCa < 4.2 mg/dL with or without symptoms of hypocalcemia, or (3) persistent requirement for calcitriol or calcium supplementation regardless of iPTH and iCa levels.

This study was approved by the E-Da Hospital Institutional Review Board (EMRP-113-044/ed112358). SPSS (version 18.0 for Windows; SPSS Inc., Chicago, IL, USA) was used to calculate percentages, means, and standard deviations, and variables were analyzed using the chi-square test. A two-tailed *p*-value of less than 0.05 was considered statistically significant.

## 3. Results

A total of 528 patients were included in this study. This study also included two patients (one with anaplastic cancer and one with primary squamous cell carcinoma) who underwent TT and total laryngectomy due to laryngo-tracheal invasion. The demographics and clinical characteristics of all patients are shown in Table 1.

Among the 528 patients, four groups of patients were divided according to the status of intraoperative PG management: group 1, 279 (52.8%) patients had at least one viable PG preserved in situ and at least one PG autotransplanted; group 2, 183 (34.7%) patients had at least one viable PG preserved in situ, but no PG autotransplanted; group 3, 16 (3.0%) patients had no viable PG preserved in situ, but at least one PG autotransplanted; and group 4, 50 (9.5%) patients had no visible PG preserved in situ or autotransplanted.

At PO-1D, 434 patients (82.2%) had iPTH ≥ 15 pg/mL, 65 (12.3%) patients had iPTH between 4 and <15 pg/mL, and 29 (5.5%) patients had iPTH < 4 pg/mL (undetectable). At PO-12M, 527 patients (99.81%) had iPTH ≥ 15 pg/mL, 1 patient (0.19%) had iPTH between 4 and <15 pg/mL, and 0 patients had iPTH < 4 pg/mL. The distribution of iPTH levels during each postoperative period (PO-2W, 2M, 6M, and 12M) is shown in Table 2.

Five (0.95%) patients developed PHPS due to the continued need for calcitriol or calcium supplementation after PO-12M. Long-term iPTH follow-up of the five patients with PHPS showed that four patients had iPTH ≥ 15 pg/mL and one patient had iPTH between 4 and <15 pg/mL. The clinical characteristics and detailed iPTH and iCa follow-up of these patients are summarized in Table 3.

The incidence of PHPS showed significant difference between the 462 patients (groups 1 and 2), with at least one viable PG preserved in situ and 66 patients (groups 3 and 4) without viable PG preserved in situ (0.2% vs. 6.1%, *p* < 0.001). In addition, the results show the proportion of patients in each group who underwent CND and LND. The results indicate that patients with no visible PG preserved in situ had a lower proportion of LND (*p* = 0.128) and a significantly lower proportion of CND (*p* = 0.005) compared to those with at least one visible PG preserved in situ. These findings are summarized in Table 4.

All patients were discharged at PO-2D. None of the patients presented to the emergency department, were readmitted to the hospital, or had a prolonged hospital stay due to hypocalcemia.

Eighteen patients (90%) had papillary thyroid cancer, thirteen had the classic type, four had the follicular variant, and one had the sclerotic variant; two patients (10%) had follicular thyroid cancer. *BRAF* genetic analysis was available for 10 patients; 3 (30%) were positive and 7 (70%) were negative. The postoperative lymph node metastasis status was N0, N1a, and N1b in 4, 10, and 6 patients, respectively. Indeed, aggressive growth of pediatric thyroid cancer at this institution has been demonstrated.

## 4. Discussion

Most thyroid surgeons usually rely on a color change in the exposed PG to determine whether it is viable or not, and then decide to proceed with in situ preservation or autotransplantation. However, several studies have reported that observing PG color changes to assess its vascular supply is unreliable [20,21,22]. Figure 3 shows that the PG resected from the surgical field still has normal color with fresh blood vessels on the PG capsule. Sitges-Serra [31] reviewed three studies [18,32,33] in which patients had three PGs preserved in situ, and the fourth one was autotransplanted or resected, with the incidence of permanent hypoparathyroidism ranging from 3.5% to 7.3%. De León-Ballesteros et al. [33] reported that permanent hypoparathyroidism was significantly associated with the number of PGs preserved in situ, with an incidence of 2.5% in 4 glands, 3.8% in 3 glands, and 13.3% in 1–2 glands among 1018 patients who underwent TT. They concluded that in situ preservation of at least three PGs was associated with a lower rate (2.79%) of permanent hypoparathyroidism. The results of these studies suggest that some of the PGs preserved in situ may have lost their vascular supply and may not be alive. Therefore, before deciding to preserve a PG in situ, it is necessary to ensure that the PG has a vascular supply.

To evaluate the PG vascular supply during thyroidectomy, Kuhel and Carew [20] performed an incisional biopsy of the PG to determine whether the vascular supply was intact, thereby facilitating the identification of devascularized PGs that can be salvaged with autotransplantation. The prick test is mentioned in the literature [34,35] as a method to assess PG viability. Continuous bright red bleeding after the prick test is considered a reliable indicator of adequate blood flow, while color or tissue fullness alone are not regarded as reliable indicators. Ji et al. [21] made a small incision with a cold knife to evaluate vascular supply to the PG when its vascular pedicle seemed to be unsafe. Wu et al. [22] used the fine-needle pricking test to visualize PG vascularization. Their results showed that a PG preserved in situ with excellent vascularity ruled out the possibility of hypoparathyroidism.

When performing the stabbing test, a needle or blade can be used. In this study, we routinely assessed the vascular supply of exposed PGs by stabbing test with a 23G needle. If fresh blood oozed after stabbing the gland, the PG was considered viable and preserved in situ. Even if the PG color became dark and black, we preserved it in situ after a positive stabbing test (Figure 4A,B). Promberger et al. [23] also did not support autotransplantation as a generally applicable first-line intervention for discolored PGs in the absence of other criteria for autotransplantation, claiming that the function of discolored PGs is only transiently impaired and recovers within a short time after surgery. The PG is considered non-viable if no fresh blood is oozing out after repeated stabbing, or if there is only old, congested bleeding without fresh blood. The PG was then cut piece by piece from the distal end to confirm the complete loss of vascular supply and was autotransplanted into the muscle pocket. With this PG management strategy, 434 patients (82.2%) had iPTH ≥ 15 pg/mL, 65 patients (12.3%) had iPTH between 4 and <15 pg/mL, and 29 patients (5.5%) had iPTH < 4 pg/mL at PO-D1. At PO-12M, 527 patients (99.81%) had iPTH ≥ 15 pg/mL, 1 patient (0.19%) had iPTH between 4 and <15 pg/mL, and 0 patients had iPTH < 4 pg/mL. Although five patients (0.95%) developed PG insufficiency and required long-term use of calcitriol or calcium supplementation, the results can be considered as a good long-term outcome of PG function compared to the literature. Several studies with respect to the application of ICG angiography also showed good results in patients, with one well-perfused PG preserved in situ [25,26,27]. However, compared with ICG angiography, the PG stabbing test is a simpler, more useful, and more cost-effective way to assess PG vascular supply and viability. The new technology of near-infrared autofluorescence may be valuable and helpful for PG identification and preservation [24,36]. In addition, a thorough understanding of PG anatomy and surgical techniques is also important for preserving PG function [37].

In this study, we found instructive evidence of the association between intraoperative PG management status and permanent hypoparathyroidism. The incidence of PHPS was 0.2% in the 462 patients with at least one viable PG preserved in situ following the positive stabbing, and 6.1% in the 66 patients without a viable PG preserved in situ. This shows a significant difference. The results of this study indicate that the preservation of at least one viable PG in situ is extremely important in terms of preventing the occurrence of PHPS. Analysis of the CND/LND proportions indicates that the higher incidence of PHPS in patients with no visible PG preserved in situ is not due to the performance of CND or LND. The underlying mechanism requires further investigation. Figure 5 shows a primary thyroid squamous cell carcinoma with laryngotracheal invasion. After TT and total laryngectomy, only one PG was found and preserved with a positive stabbing test. The postoperative iPTH level decreased significantly from 59 pg/mL to 4.2 pg/mL at PO-1D, but recovered quickly to 19.4 pg/mL at PO-2W and 25.4 pg/mL at PO-2M.

There are several limitations in this study:(1)The intraoperative PG management status of this study focuses on the exposed PGs, as identifying all PGs and assessing their vascular supply in every PG is not feasible. Invisible PGs may remain embedded in the thyroid bed with variable vascularity or be inadvertently resected.(2)Quantitative evaluation of PG vascularity is challenging. Active blood oozing after the stabbing test suggests good vascularization, while absent or minimal oozing indicates poor vascularity. If blood continues to ooze after wiping with gauze, it indicates that the PG is partially vascularized and can be preserved in situ.(3)The stabbing test is an invasive procedure that may cause trauma to the PGs, though our findings suggest minimal impact on function.(4)This is a single-center, single-surgeon study, which may limit generalizability. While standardized surgical and postoperative management protocols were followed, multicenter validation is needed.(5)The study focuses on postoperative hypoparathyroidism; other complications such as vocal fold paralysis were not analyzed. However, we acknowledge their clinical importance and will continue to monitor these issues.(6)Intraoperative PTH (ioPTH) assays were not included, as our focus was on long-term outcomes rather than intraoperative fluctuations. While this may limit real-time assessment of parathyroid function, its impact is minimal given the study’s emphasis on long-term recovery. Incorporating ioPTH could provide additional insights into early parathyroid function.(7)The underlying thyroid disease may influence postoperative calcium and parathyroid function, but as this study focuses on intraoperative strategies, we did not analyze this correlation. Further studies could explore its impact on hypocalcemia and hypoparathyroidism prevention.(8)Due to institutional and practical limitations, we did not routinely assess phosphorus levels, multivitamin intake, vitamin D monitoring, or a 3-day calcium intake record, which limited preoperative evaluation and postoperative analysis. We acknowledge that these factors may influence the study results, and a more comprehensive understanding of their impact on long-term postoperative parathyroid function would require further investigation in future studies.

## 5. Conclusions

Permanent hypoparathyroidism is an extremely important issue when performing TT. From the results of this study, PHPS is rare when ensuring at least one viable PG is preserved in situ. In this study, a PG was not found in patients who were at high risk of PHPS. The unidentified PG may have been inadvertently removed or the vascular supply may have been completely or partially lost. In addition, the stabbing test is a simple, useful, and cost-effective method to assess the vascular supply of exposed PGs, providing surgeons with the essential information for intraoperative PG management.

## Figures and Tables

**Figure 1 diagnostics-15-00593-f001:**
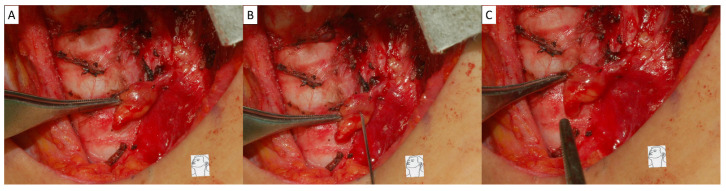
Positive stabbing test. (**A**) The parathyroid gland (PG) was meticulously dissected from the thyroid capsule with vessel pedicle preserved. (**B**) A stabbing test was performed to assess the vascular supply. (**C**) If fresh blood oozing was found after stabbing, the PG was considered viable and preserved in situ. The symbol represents the cranial-to-caudal orientation of the surgery.

**Figure 2 diagnostics-15-00593-f002:**
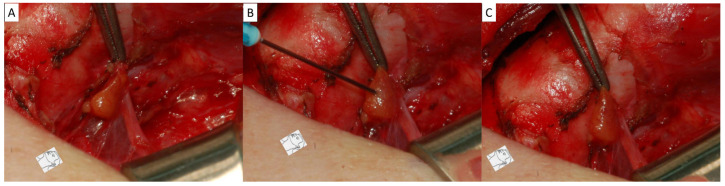
Negative stabbing test. (**A**) The PG was identified with vessel pedicle. (**B**) Stabbing test to assess the vascular supply. (**C**) Since there was no fresh blood oozing after repeated PG stabbing, the PG was considered non-viable and autotransplanted into the muscle pocket.

**Figure 3 diagnostics-15-00593-f003:**
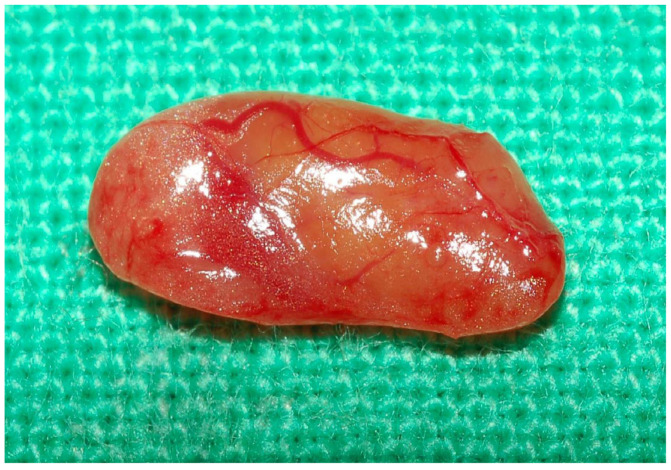
The devascularized PG still has normal color with fresh blood vessels on the PG capsule.

**Figure 4 diagnostics-15-00593-f004:**
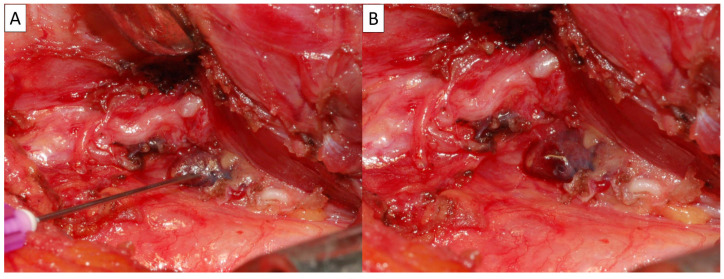
Discolored PG with positive stabbing test. (**A**) The PG has become black and dark. (**B**) Since fresh blood oozing was found after stabbing, the PG was considered viable and preserved in situ.

**Figure 5 diagnostics-15-00593-f005:**
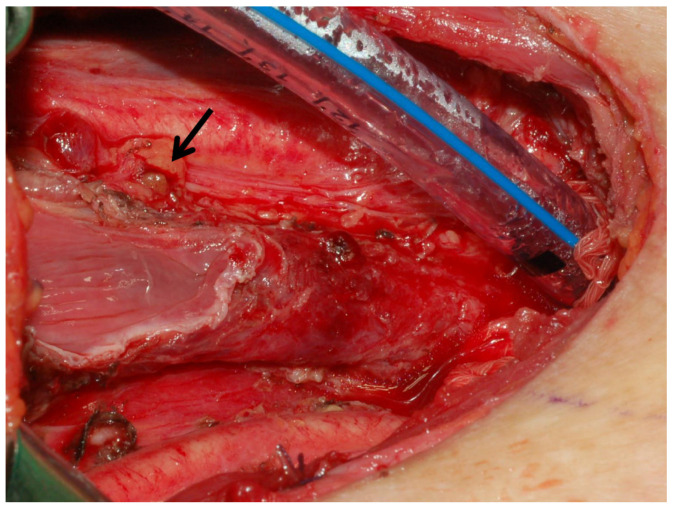
The patient underwent total thyroidectomy and total laryngectomy. Only one PG (↑) following a positive stabbing test was preserved in situ. Transient postoperative hypoparathyroidism occurred, but the patient recovered quickly during subsequent follow-up.

**Table 1 diagnostics-15-00593-t001:** The demographics and clinical characteristics of the patients.

Case number		528
Age (year, Mean ± SD)		51.8 ± 12.2
Sex		
	Male	82 (15.5%)
	Female	446 (84.5%)
Pathologic report		
	Benign	251 (47.5%)
	Malignant	277 (52.5%)
Central Neck Dissection		
	Without	308 (58.3%)
	With	220 (41.7%)
	Unilateral	193 (36.6%)
	Bilateral	27 (5.1%)
Lateral Neck Dissection		
	Without	493 (93.4%)
	With	35 (6.6%)
	Unilateral	32 (6.0%)
	Bilateral	3 (0.6%)
Intraoperative PG management status		
	(1) At least one PG preserved in situ and at least one PG autotransplanted	279 (52.8%)
	(2) At least one PG preserved in situ, but no PG autotransplanted	183 (34.7%)
	(3) No viable PG preserved in situ, but at least one PG autotransplanted	16 (3.0%)
	(4) No visible PG preserved in situ or autotransplanted	50 (9.5%)

iPTH = intact parathyroid hormone; PG = parathyroid gland; SD = standard deviation.

**Table 2 diagnostics-15-00593-t002:** Distribution of iPTH levels during each postoperative period among 528 patients.

iPTH Level (pg/mL)	≧15	≧4 and <15	<4
PO-1D	434 (82.2%)	65 (12.3%)	29 (5.5%)
PO-2W	459 (86.9%)	43 (8.1%)	26 (4.9%)
PO-2M	505 (95.6%)	21 (4.0%)	2 (0.4%)
PO-6M	523 (99.05%)	5 (0.95%)	0 (0.0%)
PO-12M	527 (99.81%)	1 (0.19%)	0 (0.0%)

iPTH = intact parathyroid hormone; PO = postoperative.

**Table 3 diagnostics-15-00593-t003:** Characteristics and long-term iPTH amd iCa follow-up of five patients with PHPS.

Case Number	PP	PA	B/M	CND		Pre-op	PO-1D	PO-2W	PO-2M	PO-6M	PO-12M
1	2	0	B	−	iPTH	85.4	<4.0	5.6	9.4	21.6	16.0
					iCa	4.44	3.85	4.52	4.44	3.90	4.64
2	0	2	M	−	iPTH	78.3	<4.0	<4.0	8.5	12.6	18.0
					iCa	4.58	3.73	5.33	4.41	4.49	4.45
3	0	0	M	+	iPTH	54.5	7.2	<4.0	22.3	18.3	17.5
					iCa	4.60	4.22	4.63	3.84	3.97	3.93
4	0	0	B	+	iPTH	53.2	<4.0	<4.0	15.1	17.1	18.2
					iCa	4.57	4.28	4.88	4.62	4.56	4.33
5	0	0	B	−	iPTH	21.5	<4.0	<4.0	5.5	7.6	11.8
					iCa	4.77	4.31	4.83	4.10	4.25	4.30

iPTH = intact parathyroid hormone (pg/mL); iCa = ionized calcium (mg/dL); PHPS = permanent hypoparathyroidism status; PP = parathyroid gland (PG) preservation; PA = PG autotransplantation; B = benign; M = malignant; CND = central neck dissection; + indicates performed, − indicates not performed; Pre-op = preoperative; PO = postoperative.

**Table 4 diagnostics-15-00593-t004:** Correlation between intraoperative PG management status and permanent hypoparathyroidism.

Intraoperative Status of PG Management		Case Number	CND	LND	PHPS
**At least one visible PG preserved in situ**		**462**	**203 (46.9%)**	**34 (7.4%)**	**1 (0.2%)**
1.	At least one PG preserved in situ and at least one PG autotransplanted	279	147	23	0 (0.0%)
2.	At least one PG preserved in situ, but no PG autotransplanted	183	56	11	1 (0.5%)
**No visible PG preserved in situ**		**66**	**17 (25.8%)**	**1 (1.5%)**	**4 (6.1%)**
3.	No visible PG preserved in situ, but at least one PG autotransplanted	16	8	1	1 (6.3%)
4.	No visible PG preserved in situ or autotransplanted	50	9	0	3 (6.0%)
**At least one visible PG preserved in situ** vs. **no visible PG preserved in situ**			***p* = 0.005**	***p* = 0.128**	***p* < 0.001**

PG = parathyroid gland; CND = central neck dissection; LND = lateral neck dissection; HP = hypoparathyroidism.

## Data Availability

The original contributions presented in the study are included in the article. Further inquiries can be directed to the corresponding authors.

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
