# Peer review of "Intraoperative Management of Parathyroid Glands and Long-Term Outcome of Parathyroid Function Following Total Thyroidectomy"

_diagnostics, 2025, doi:10.3390/diagnostics15050593_

Round 1
Reviewer 1 Report
Comments and Suggestions for Authors
Dear Authors,
The article is interesting.
Here are my points:
1. Title. Please use “outcome” instead of “outcomes”.
2. Title. Please use “after” or “post” or “following” instead of “in total thyroidectomy”.
3. Title. Please remove “patients” since this is obvious. Moreover, “total thyroidectomy patients” is not adequate (e.g. alternatively, “patients who underwent total thyroidectomy” or “surgery candidates” depending on the specific statement).
4. First page needs re-adjustments since it is blanc.
5. Abstract. Background. Auto-transplantation of the parathyroid glands is not a common practice amid thyroidectomy and it dependents of the underlying endocrine ailment and the surgical practice of one center, rather after certain types of parathyroidectomy.
6. Abstract. Please “parathyroid glands” since there are at least 4 (usually).
7. Abstract. The first sub-section of the Methods it should the study design followed by inclusion and exclusion criteria of the studied population. The actual number of the enrolled patients is obtained after applying the inclusion/exclusion criteria and it should be the first data amid Results section. Please correct the same aspects within the main text.
8. Abstract…“continued need for supplementation” = you need to mention “supplementation of calcium and vitamin D” since otherwise this is a thyroidectomy, and levothyroxine should also be supplemented.
9. Line 52, page “4-<15pg/mL” = do you mean “between 4 and 15 pg/mL”?
10. “POD 12 months” = this is not clear since “POD” means post-op DAY, and then you added months
11. Keywords: please avoid capital letters
12. Introduction. You cited studies with one third of the patients displaying post-operatory hypoparathyroidism which is too much, and it does not represent the current (general) results.
13. In Introduction I suggest using or mentioning the data from current guidelines with respect to auto-transplantation and the rate of post-thyroidectomy hypoparathyroidism.
14. Objective. This needs to be a simple, clear, reproducible statement (no references or justification in this objective statement)
15. Methods. The subsection describing the Patients should be at Results or at least after you provide the general information about the study and the studied population.
16. Figure 1. I could not find it.
17. Figure 2. Is this the same patient? The captures are from the authors’ experience? The same questions are for Figure 3 and 4.
18. The tables need to respect the MDPI rules.
19. Results. Line 203, page 7. Please start by providing the actual information and then a reference to the Table. The text provides the data and it does not represent a supplementary panel to the table.
20. Discussion. Limitations. Do you have any data with respect to synchronous side effects (other than hypoparathyroidism) e.g. vocal cords injury?
21. Limitations. Lack of intra-operatory PTH assays (not after surgery) – is it a potential bias?
22. Limitations. This is a single center experience. Do you think it represents a limit of the study?
23. Discussion. Do you consider that a correlation with the underlying thyroid disease in surgery candidates might help to forestall post-operatory hypocalcemia and/or hypoparathyroidism?
Well done!
Thank you
Author Response
Comment
- Title. Please use “outcome” instead of “outcomes”.
- Title. Please use “after” or “post” or “following” instead of “in total thyroidectomy”.
- Title. Please remove “patients” since this is obvious. Moreover, “total thyroidectomy patients” is not adequate (e.g. alternatively, “patients who underwent total thyroidectomy” or “surgery candidates” depending on the specific statement).
Response 1-3. We have revised the title to “Intraoperative Management of Parathyroid Glands and Long-term Outcome of Parathyroid Function Following Total Thyroidectomy” Thank you for your suggestion.
Comment
- First page needs re-adjustments since it is blanc.
- Abstract. Background. Auto-transplantation of the parathyroid glands is not a common practice amid thyroidectomy and it dependents of the underlying endocrine ailment and the surgical practice of one center, rather after certain types of parathyroidectomy.
- Abstract. Please “parathyroid glands” since there are at least 4 (usually).
Response 4-6. Thank you for the suggestions. We have made the corresponding revisions in the relevant sections.
Comment
- Abstract. The first sub-section of the Methods it should the study design followed by inclusion and exclusion criteria of the studied population. The actual number of the enrolled patients is obtained after applying the inclusion/exclusion criteria and it should be the first data amid Results section. Please correct the same aspects within the main text.
- Abstract…“continued need for supplementation” = you need to mention “supplementation of calcium and vitamin D” since otherwise this is a thyroidectomy, and levothyroxine should also be supplemented.
- Line 52, page “4-<15pg/mL” = do you mean “between 4 and 15 pg/mL”?
Response 7-9. The abstract has been revised according to the suggestions, and the main text has been reviewed and modified accordingly.
Comment 10. “POD 12 months” = this is not clear since “POD” means post-op DAY, and then you added months
Response 10. All relevant descriptions in the text have been revised to PO-1D, PO-2W, PO-2M, PO-6M, and PO-12M to minimize potential misunderstandings for readers.
Comment 11. Keywords: please avoid capital letters
Response 11. Revised
Comment
- Introduction. You cited studies with one third of the patients displaying post-operatory hypoparathyroidism which is too much, and it does not represent the current (general) results.
- In Introduction I suggest using or mentioning the data from current guidelines with respect to auto-transplantation and the rate of post-thyroidectomy hypoparathyroidism.
Response 12-13.
We cited a meta-analysis published in 2024 and selected data from studies conducted between 2021 and 2024 to enhance the discussion in our article, aiming to provide readers with a clearer understanding of the most up-to-date data.
Comment
- Objective. This needs to be a simple, clear, reproducible statement (no references or justification in this objective statement)
- Methods. The subsection describing the Patients should be at Results or at least after you provide the general information about the study and the studied population.
Response 14-15. The paragraph has been revised according to your suggestion.
Comment 16. Figure 1. I could not find it.
Response 16. It may have been a formatting error. You can now find the image in the revised version
Comment 17. Figure 2. Is this the same patient? The captures are from the authors’ experience? The same questions are for Figure 3 and 4.
Response 17. The three images in Figure 2 are from the same patient. Parathyroid gland identification relies on clinical experience, whereas the stabbing test is more objective. Figures 3 and 4 each represent different patients.
Comment
- The tables need to respect the MDPI rules.
- Results. Line 203, page 7. Please start by providing the actual information and then a reference to the Table. The text provides the data and it does not represent a supplementary panel to the table.
Response 18-19. The manuscript has been revised according to your suggestion.
Comment 20. Discussion. Limitations. Do you have any data with respect to synchronous side effects (other than hypoparathyroidism) e.g. vocal cords injury?
Response 20. Thank you for your insightful question. While vocal cord injury and other synchronous side effects are important considerations in thyroid surgery, this study specifically focuses on postoperative hypoparathyroidism and its related factors. As a result, data on other complications, including vocal cord injury, were not included in our analysis. We added Limitation (5) to state this issue. We recognized the clinical significance of these complications and will continue to monitor this issue in future research.
Comment
- Limitations. Lack of intra-operatory PTH assays (not after surgery) – is it a potential bias?
- Limitations. This is a single center experience. Do you think it represents a limit of the study?
- Discussion. Do you consider that a correlation with the underlying thyroid disease in surgery candidates might help to forestall post-operatory hypocalcemia and/or hypoparathyroidism?
Response 21-23. Thank you very much for your suggestions. We believe these points should be addressed in the Limitations section. Please refer to Limitations (4), (6), and (7), where we have made the corresponding revisions.
We sincerely appreciate the reviewers for providing many insightful suggestions. These not only significantly improved the quality of our revised manuscript but also gave us valuable ideas for future research design. Once again, we thank the reviewers for their patience and thorough review.
Reviewer 2 Report
Comments and Suggestions for Authors
Abstract section
The abstract should be corrected and primary and secondary outcomes should be addressed.
Introduction section
The introduction provide sufficient background and purpose.
Method section
How many did these patients have vitamin D deficiency or insufficiency? Because serum PTH can increase in patients with vitamin D deficiency.
All exclusion criteria should be given for this study. (serum calcium above high range, serum phosphorus above high limit, bone disease, nutritional rickets, other rickets, multivitamin intake, ……)
As calcium intake plays important role for supressing parathormone level in the pathogenesis of rickets, it should be considered in this study. Calcium intake for 3 days food record should be given. Distribution of serum calcium levels during each postoperative period should be given among 528 patients.
Characteristics and long-term serum calcium and phosphorus levels should be given follow-up of 5 patients with PHPS.
English language is not fine. The manuscript must be revised by native English-speaking.
Author Response
Comment 1.
Abstract section
The abstract should be corrected and primary and secondary outcomes should be addressed.
Response 1. The corresponding modifications have been made in the abstract.
Comment 2.
Method section
How many did these patients have vitamin D deficiency or insufficiency? Because serum PTH can increase in patients with vitamin D deficiency.
Response 2. Due to institutional and practical limitations, we did not routinely assess vitamin D levels. We have added a statement in Limitation (8). We also acknowledge that these factors may influence the study results, and a more comprehensive understanding of their impact on long-term postoperative parathyroid function would require further investigation in future studies.
Comment 3.
All exclusion criteria should be given for this study. (serum calcium above high range, serum phosphorus above high limit, bone disease, nutritional rickets, other rickets, multivitamin intake, ……)
Response 3. We agree and have revised the description of the Methods/exclusion criteria (lines 110-111). However, since we were unable to address for phosphorus levels and multivitamin intake, we have included these limitations in Limitation (8).
Comment 4.
As calcium intake plays important role for supressing parathormone level in the pathogenesis of rickets, it should be considered in this study.
Response 4. Thank you for your comment. This is why we used the PHPS, a more stringent definition that not only considers whether iPTH is within the normal range but also classifies patients as abnormal if they have low iCa or require ongoing calcium or vitamin D supplementation.
Comment 5.
Calcium intake for 3 days food record should be given. Distribution of serum calcium levels during each postoperative period should be given among 528 patients. Characteristics and long-term serum calcium and phosphorus levels should be given follow-up of 5 patients with PHPS.
Response 5.
We have revised Table 3 to include the iCa data for the five patients with PHPS. While we reviewed the comprehensive iCa data for all patients, these values are significantly influenced by calcium and vitamin D supplementation, leading to considerable variability and difficult to analyze in current study. To ensure a clinically meaningful interpretation, we analyzed hypocalcemia status rather than raw numerical values. We hope readers could recognize the importance of intraoperative PG preservation status and the vascular supply and to reduce the occurrence of PHPS in patients.
Comment 6.
English language is not fine. The manuscript must be revised by native English-speaking.
Response 6.
We acknowledge our limitations in language. Following this major revision, we are more than willing to have the final version professionally edited before publication to ensure the highest reading quality for our audience.
We sincerely appreciate the reviewers for providing many insightful suggestions. These not only significantly improved the quality of our revised manuscript but also gave us valuable ideas for future research design. Once again, we thank the reviewers for their patience and thorough review.
Round 2
Reviewer 2 Report
Comments and Suggestions for Authors
Thank you for asking me to review this paper from Dr Chiang et al. , Taiwan. It is an observational study, in which the Authors reported intraoperative parathyroid gland management strategies and the long-term outcomes of parathyroid gland function in total thyroidectomy patients. The method and result sections are well organized.
The English is fine and does not require any improvement.